# Learning Enriched Features via Selective State Spaces Model for Efficient Image Deblurring

## ABSTRACT

Image deblurring aims to restore a high-quality image from its corresponding blurred. The emergence of CNNs and Transformers has enabled significant progress. However, these methods often face the dilemma between eliminating long-range degradation perturbations and maintaining computational efficiency. While the selective state space model (SSM) shows promise in modeling long-range dependencies with linear complexity, it also encounters challenges such as local pixel forgetting and channel redundancy. To address this issue, we propose an efficient image deblurring network that leverages selective state spaces model to aggregate enriched and accurate features. Specifically, we introduce an aggregate local and global information block (ALGBlock) designed to effectively capture and integrate both local invariant properties and non-local information. The ALGBlock comprises two primary modules: a module for capturing local and global features (CLGF), and a feature aggregation module (FA). The CLGF module is composed of two branches: the global branch captures long-range dependency features via a selective state spaces model, while the local branch employs simplified channel attention to model local connectivity, thereby reducing local pixel forgetting and channel redundancy. In addition, we design a FA module to accentuate the local part by recalibrating the weight during the aggregation of the two branches for restoration. Experimental results demonstrate that the proposed method outperforms state-of-the-art approaches on widely used benchmarks.

## KEYWORDS

Image deblurring, state spaces model, features aggregation

## 1 INTRODUCTION

Image deblurring aims to recover a latent sharp image from its corrupted counterpart. Due to the ill-posedness of this inverse problem, many conventional approaches [12, 18] address this by explicitly incorporating various priors or hand-crafted features to constrain the solution space to natural images. Nonetheless, designing such priors proves challenging and lacks generalizability, which are impractical for real-world scenarios.

Stimulated by the success of deep learning for high-level vision tasks, numerous data-driven methods have resorted CNN as a preferable choice and develop kinds of network architectural designs, including encoder-decoder architectures [4, 7, 9], multi-stage networks [5, 47], dual networks [2, 35, 39], generative models [22, 23, 49], and so on. While the convolution operation effectively models local connectivity, its intrinsic characteristics, such as limited local receptive fields and independence of input content, hinder the model's ability to eliminate long-range dependency features. To alleviate such limitations, various transformer variants [13, 21, 43, 45, 46] have been applied to image deblurring and have achieved better performance than the CNN-based methods as

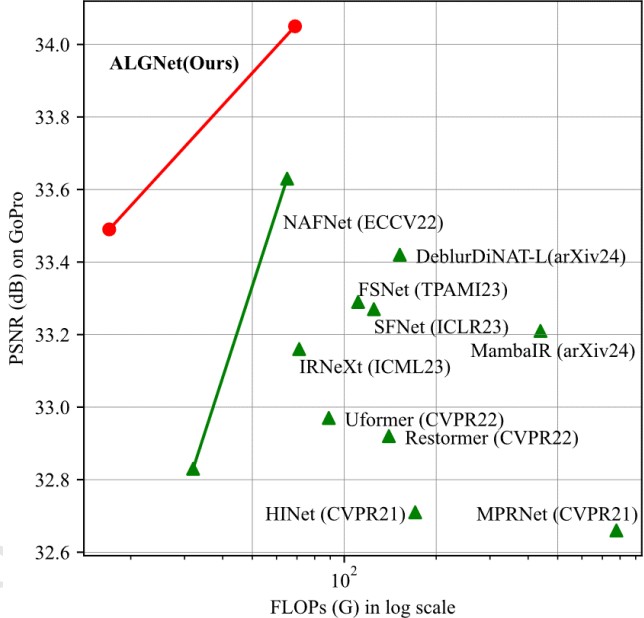

**Figure 1: Computational cost vs. PSNR of models on the Go-Pro dataset [34]. Our ALGNet achieve the SOTA performance while simultaneously reducing computational costs.**

they can better model the non-local information. However, image deblurring often deals with high-resolution images, and the attention mechanism in Transformers incurs quadratic time complexity, resulting in significant computational overhead. To alleviate computational costs, some methods [11, 46] opt to apply self-attention across channels instead of spatial dimensions. However, this approach fails to fully exploit the spatial information, which may affect the deblurring performance. While other methods [13, 45] utilize non-overlapping window-based self-attention for single image deblurring, the coarse splitting approach still falls short in fully exploring the information within each patch.

State space models [15, 33, 40], notably the enhanced version Mamba, have recently emerged as efficient frameworks due to their ability to capture long-range dependencies with linear complexity. However, Mamba's [15] recursive image sequence processing method tends to neglect local pixels, while the abundance of hidden states in the state space equation often results in channel redundancy, thereby impeding channel feature learning. Given the critical importance of local and channel features in image deblurring, directly applying the state space model often leads to poor performance. MambaIR [17] introduces the vision state space module, which utilizes a four-direction unfolding strategy to scan along four different directions for local enhancement, and incorporates channel attention to mitigate channel redundancy. Nevertheless,

this four-directional scanning approach and the computation of state spaces result in increased computational overhead, potentially sacrificing the advantage of low computational resource utilization offered by the state spaces equation.

Taking into account the above analyses, a natural question arises: Is it feasible to design a network that efficiently aggregates local and global features for image deblurring? To achieve this objective, we propose ALGNet, with several key components. Specifically, we present an aggregate local and global information block (ALGBlock) aimed at efficiently capturing and merging both local invariant properties and long-range dependencies. This ALGBlock consists of two key modules: a module dedicated to capturing local and global features (CLGF), and a feature aggregation module (FA). The CLGF module is further divided into two branches: the global branch, which utilizes a selective state spaces model to capture long-range dependency features with linear complexity, and the local branch, which incorporates simplified channel attention to effectively model local connectivity. This combination not only addresses issues like local pixel forgetting and channel redundancy but also empowers the network to capture more enriched and precise features. Additionally, given that image details are predominantly comprised of local features of images [6], we design the FA module to underscores the significance of the local information in the restoration process by dynamically recalibrating the weights through a learnable factor during the aggregation of the CLGF two branches for restoration. Finally, we implement multiple scales for both input and output modes, aiming to alleviate training difficulty. As illustrated in Figure 1, our ALGNet model achieves state-of-the-art performance while preserving computational efficiency compared to existing methods.

The main contributions of this work are:

(1) We propose ALGNet, an efficient network for aggregating enriched and precise features leveraging a selective state spaces model for image deblurring. ALGNet consists of multiple ALGBlocks, each with a capturing local and global features module (CLGF) and a feature aggregation module (FA).

(2) We design the CLGF module to capture long-range dependency features using a selective state spaces model, while employing simplified channel attention to model local connectivity, thus reducing local pixel forgetting and channel redundancy.

(3) We present the FA module to emphasize the importance of the local features in restoration by recalibrating the weights through the learnable factor.

(4) Extensive experiments demonstrate that our ALGNet achieves favorably performance against state-of-the-art methods.

## 2 RELATED WORK

### 2.1 Hand-crafted prior-based methods.

Due to the image deblurring ill-posed nature, many conventional approaches [12, 18] tackle this problem by relying on hand-crafted priors to constrain the set of plausible solutions. However, designing such priors is a challenging task and usually lead to complicated optimization problems.

### 2.2 CNN-based methods.

With the rapid advancement of deep learning, instead of manually designing image priors, lots of methods [2, 4, 22, 23, 35, 39, 47, 49] develop kinds of deep CNNs to solve image deblurring. To better explore the balance between spatial details and contextualized information, MPRNet [47] propose a cross-stage feature fusion to explore the features from different stages. MIRNet-V2 [48] introduces a multi-scale architecture to learn enriched features for image restoration. IRNeXt [9] rethink the convolutional network design and exploit an efficient and effective image restoration architecture based on CNNs. NAFNet [4] analyze the baseline modules and presents a simplified baseline network by either removing or replacing nonlinear activation functions. SFNet [10] and FSNet [8] design a multi-branch dynamic selective frequency module and a multi-branch compact selective frequency module to dynamically select the most informative components for image restoration. Although these methods achieve better performance than the hand-crafted prior-based ones, the intrinsic properties of convolutional operations, such as local receptive fields, constrain the models' capability to efficiently eliminate long-range degradation perturbations.

### 2.3 Transformer-based methods.

Due to the content-dependent global receptive field, the transformer architecture [44] has recently gained much popularity in image restoration [3, 13, 27, 43, 45, 46, 51], demonstrating superior performance compared to previous CNN-based baselines. IPT [3] employs a Transformer-based multi-head multi-tail architecture, proposing a pre-trained model for image restoration tasks. However, image deblurring often deals with high-resolution images, and the attention mechanism in Transformers incurs quadratic time complexity, resulting in significant computational overhead. In order to reduce the computational cost, Uformer [45], SwinIR [27] and $U^2$former [13] computes self-attention based on a window. Nonetheless, the window-based approach still falls short in fully exploring the information within each patch. Restormer [46] and MRLPFNet [11] compute self-attention across channels rather than the spatial dimension, resulting in the linear complexity. However, this approach fails to fully exploit the spatial information. FFTformer [21] explores the property of the frequency domain to estimate the scaled dot-product attention, but need corresponding inverse Fourier transform, leading to additional computation overhead.

### 2.4 State Spaces Model.

State spaces models [15, 16, 32, 33, 40, 41] have recently emerged as efficient frameworks due to their ability to capture long-range dependencies with linear complexity. S4 [16] is the first structured SSM to model long-range dependency. S5 [41] propose the diagonal SSM approximation and computed recurrently with the parallel scan. Mega [32] introduced a simplification of S4 [16] to be real- instead of complex- valued, giving it an interpretation of being an exponential moving average. SGConv [14] and LongConv [26] focus on the convolutional representation of S4 and create global or long convolution kernels with different parameterizations. Mamba [15] propose a selective mechanism and hardware-aware parallel algorithm. Many vision tasks start to employ Mamba to tackle image

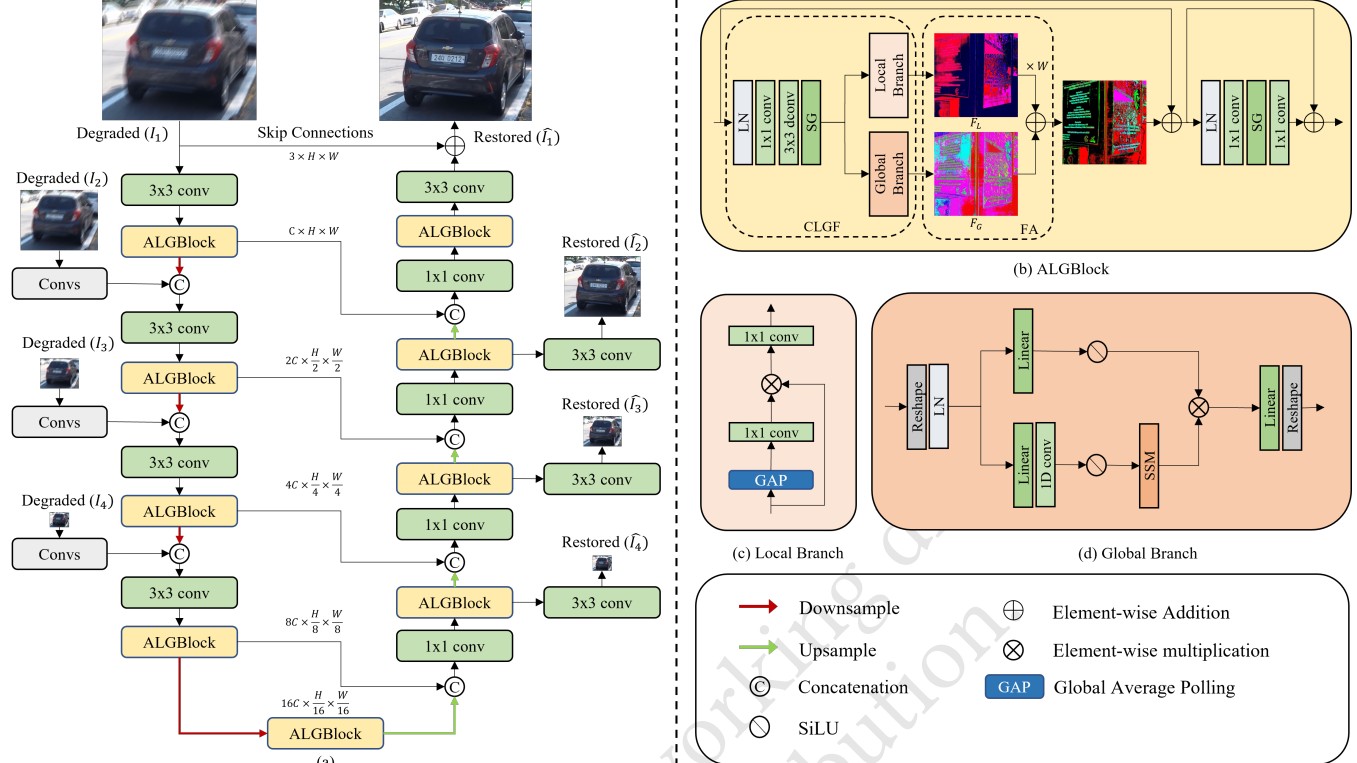

**Figure 2: Overall architecture of ALGNet. (a) ALGNet consists of several ALGBlocks and adopts the multi-input and multi-output strategies for image restoration. (b) ALGBlock comprises two primary modules: a module for capturing local and global features (CLGF), and a feature aggregation module (FA). The CLGF module is composed of two branches: (c) the local branch to model local connectivity, while (d) the global branch captures long-range dependency features.**

classification [29, 52], image segmentation [31], and so on. However, the standard Mamba model still encounters issues with local pixel forgetting and channel redundancy when applied to image restoration tasks. To tackle this challenge, MambaIR [17] adopts a four-direction unfolding strategy to scan along four different directions and integrates channel attention. Nevertheless, this four-directional scanning approach results in increased computational overhead. In this work, we design an ALGBlock to efficiently capture and integrate both local invariant properties and long-range dependencies through a selective state spaces model with lower computational cost. As depicted in Figure 1, our ALGNet outperforms MambaIR [17] while reducing computational costs by 96.1%.

## 3 METHOD

In this section, we first outline the overall pipeline of our ALGNet. Subsequently, we delve into the details of the proposed ALGBlock, which includes the capturing local and global features module (CLGF) and the feature aggregation module (FA).

### 3.1 Overall Pipeline

The overall pipeline of our proposed ALGNet, shown in Figure 2, adopts a single U-shaped architecture for image deblurring. Given

a degraded image $\mathbf{I} \in \mathbb{R}^{H \times W \times 3}$, ALGNet initially applies a convolution to acquire shallow features $\mathbf{F_0} \in \mathbb{R}^{H \times W \times C}$ ($H, W, C$ are the feature map height, width, and channel number, respectively). These shallow features undergo a four-scale encoder sub-network, progressively decreasing resolution while expanding channels. It's essential to note the use of multi-input and multi-output mechanisms for improved training. The low-resolution degraded images are incorporated into the main path through the Convs (consists of multiple convolutions and ReLU) and concatenation, followed by convolution to adjust channels. The in-depth features then enter a middle block, and the resulting deepest features feed into a four-scale decoder, gradually restoring features to the original size. During this process, the encoder features are concatenated with the decoder features to facilitate the reconstruction. Finally, we refine features to generate residual image $\mathbf{X} \in \mathbb{R}^{H \times W \times 3}$ to which degraded image is added to obtain the restored image: $\hat{\mathbf{I}} = \mathbf{X} + \mathbf{I}$. It's important to note that the three low-resolution results are solely used for training.

We optimize the proposed network ALGNet with the following loss function:

$$L = \sum_{i=1}^{4} (L_{char}(\hat{I}_i, \bar{I}_i) + \delta L_{edge}(\hat{I}_i, \bar{I}_i) + \lambda L_{freq}(\hat{I}_i, \bar{I}_i)) \quad (1)$$

Anonymous Authors

where $i$ denotes the index of input/output images at different scales, $\bar{I}_i$ denotes the target images and $L_{char}$ is the Charbonnier loss:

$$L_{char} = \sqrt{||\hat{I}_i - \bar{I}_i||^2 + \epsilon^2} \qquad (2)$$

with constant $\epsilon$ empirically set to 0.001 for all the experiments. $L_{edge}$ is the edge loss:

$$L_{edge} = \sqrt{||\triangle \hat{I}_i - \triangle \bar{I}_i||^2 + \epsilon^2} \qquad (3)$$

where $\triangle$ represents the Laplacian operator. $L_{freq}$ denotes the frequency domains loss:

$$L_{freq} = ||\mathcal{F}(\hat{I}_i) - \mathcal{F}(\bar{I}_i)||_1 \qquad (4)$$

where $\mathcal{F}$ represents fast Fourier transform, and the parameters $\lambda$ and $\delta$ control the relative importance of loss terms, which are set to 0.1 and 0.05 as in [8, 47], respectively.

## 3.2 Capturing Local and Global Features Module (CLGF)

Transformer-based models [11, 13, 21, 45, 46] address the limitations of CNNs, such as a limited receptive field and lack of adaptability to input content. They excel in modeling non-local information, leading to high-quality image reconstruction, and have emerged as the dominant method for image deblurring. However, image deblurring commonly involves processing high-resolution images, and the attention mechanism in Transformers introduces quadratic time complexity, leading to considerable computational overhead. While it's possible to mitigate computational consumption by utilizing window-based attention [13, 45] or channel-wise attention [11, 46], these methods inevitably lead to information loss.

To address this challenge, we design the capturing local and global features module (CLGF) depicted in Figure 2(b), aiming to capture long-range dependency features and model local connectivity with linear complexity. Specifically, given an input tensor $X_{l-1}$, we initially process it through Layer Normalization (LN), Convolution, and Simple Gate (SG) to obtain spatial features $X_{l-1}^s$ as follows:

$$X_{l-1}^s = SG(f_{3\times3}^{dwc}(f_{1\times1}^c(LN(X_{l-1})))) \qquad (5)$$
$$SG(X_{f0}) = X_{f1} \otimes X_{f2}$$

where $f_{3\times3}^{dwc}$ denotes the $3 \times 3$ depth-wise convolution, $f_{1\times1}^c$ represents $1 \times 1$ convolution. $SG(\cdot)$ is the simple gate, employed as a replacement for the nonlinear activation function. For a given input $X_{f0} \in \mathbb{R}^{H\times W\times C}$, SG initially splits it into two features $X_{f1} \in \mathbb{R}^{H\times W\times \frac{C}{2}}$ and $X_{f2} \in \mathbb{R}^{H\times W\times \frac{C}{2}}$ along channel dimension. Subsequently, SG calculates the $X_{f1}, X_{f2}$ using a linear gate. Next, we feed the spatial features $X_{l-1}^s$ through both the global branch and the local branch to capture global features $F_G$ and local features $F_L$, respectively.

In the global branch, depicted in Figure 2(c), we opt for a state-space model (SSM) instead of Transformers to capture long-distance dependencies, ensuring linear complexity. Specifically, starting with an input feature $X_{l-1}^s$, we first reshape and normalize it using layer normalization (LN). Subsequently, it undergoes processing through two parallel branches. In the top branch, the feature channels are expanded by a linear layer, followed by activation through the SiLU

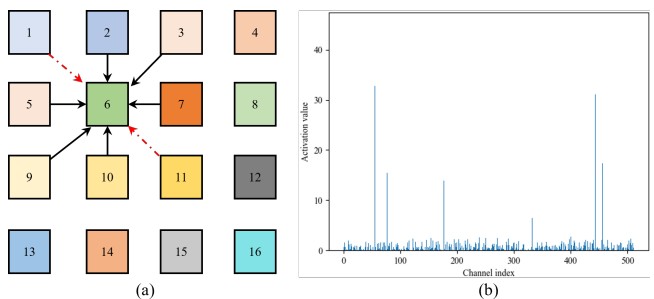

(a)         (b)

**Figure 3: (a) Local pixels (highlighted by the red dashed line) are susceptible to being forgotten in the flattened 1D sequence due to the extensive distance. (b) Following [17], we apply ReLU and global average pooling to the outputs of the global branch to obtain channel activation values. However, a considerable portion of channels remain inactive, indicating channel redundancy.**

function. In the bottom branch, the feature channels are expanded by a linear layer followed by the SiLU activation function, along with the selective state spaces model layer. The SSM inspired by the particular continuous system that maps a 1-dimensional function or sequence $x(t) \in \mathbb{R} \to y(t) \in \mathbb{R}$ through an implicit latent state $h(t) \in \mathbb{R}^N$ as follows:

$$h'(t) = \mathbf{A}h(t) + \mathbf{B}x(t)$$
$$y(t) = \mathbf{C}h(t) \qquad (6)$$

where $\mathbf{A} \in \mathbb{R}^{N\times N}, \mathbf{B} \in \mathbb{R}^{N\times 1}, \mathbf{C} \in \mathbb{R}^{1\times N}$ are four parameters, and $N$ is the state size. SSM first transform the continuous parameters $\mathbf{A, B}$ to discrete parameters $\overline{\mathbf{A}}, \overline{\mathbf{B}}$ through fixed formulas $\overline{\mathbf{A}} = exp(\square\mathbf{A})$ and $\overline{\mathbf{B}} = (\square\mathbf{A})^{-1}exp(\square\mathbf{A} - I) \cdot \square\mathbf{B}$, where $\square$ denotes the timescale parameter. After the discretization, the model can be computed as a linear recurrence way:

$$h_t = \overline{\mathbf{A}}h_{t-1} + \overline{\mathbf{B}}x_t$$
$$y_t = \mathbf{C}h_t \qquad (7)$$

or a global convolution way:

$$\overline{\mathbf{K}} = (\mathbf{C}\overline{\mathbf{B}}, \mathbf{C}\overline{\mathbf{A}}\overline{\mathbf{B}}, ..., \mathbf{C}\overline{\mathbf{A}}^{k-1}\overline{\mathbf{B}})$$
$$y = x \otimes \overline{\mathbf{K}} \qquad (8)$$

where $k$ is the length of the input sequence, $\otimes$ denotes convolution operation, and $\overline{\mathbf{K}} \in \mathbb{R}^k$ is a structured convolution kernel. Selective SSM integrates a selection mechanism into SSM, making the parameters input-dependent. The selective SSM offers two key advantages. Firstly, it shares the same recursive form as Eq.7, enabling the model to capture long-range dependencies to aid in restoration. Secondly, the parallel scan algorithm enables SSM to leverage the advantages of parallel processing described in Eq.8, thereby facilitating efficient training.

After that, features from the two branches are aggregated with the element-wise multiplication. Finally, the channel number is projected back and reshape to the original size. The total process

 

can be defined as:

$$F_t = SiLU(Linear(Reshape(LN(X_{l-1}^s)))),$$

$$F_b = SSM(SiLU(f_{1\times1}^c(Linear(Reshape(LN(X_{l-1}^s)))))) \quad (9)$$

$$F_G = Reshape(Linear(F_t \otimes F_b))$$

Despite the selective SSM's ability to capture long-range dependencies with linear computational complexity, it can result in issues like local pixel forgetting and channel redundancy, primarily due to the flattening strategy and an excessive number of hidden states. As shown in Figure 3(a), when the 2D feature map is flattened into a 1D sequence, adjacent pixels (e.g., sequence number 1 and 11) become widely separated, leading to the issue of pixel forgetting. Additionally, as per [17], we visualize the activation results for different channels in Figure 3(b) and observe significant channel redundancy attributed to the larger number of hidden states in the selective SSM. To address the above challenges, we equip our CLGF with a local branch to model local connectivity and facilitate the expressive power of different channels. The local branch shown in Figure 2(c), it is a simplified channel attention. Given the spatial features $X_{l-1}^s$, the local features $F_L$ can be obtained by:

$$F_L' = X_{l-1}^s \otimes f_{1\times1}^c(GAP(X_{l-1}^s)),$$

$$F_L = f_{1\times1}^c(F_L') \quad (10)$$

where GAP is the global average pool. Noted that, in our CLGF module, we initially capture the spatial feature $X_{l-1}^s$, which aggregates neighboring information, before feeding it to the global branch. This approach effectively reduces the problem of local pixel forgetting.

## 3.3 Feature Aggregation Module (FA)

Given that image details primarily consist of local features, we design a feature aggregation (FA) module, depicted in Figure 2(b), to highlight the significance of the local block in restoration. This is achieved by dynamically recalibrating the weights through a learnable factor during the aggregation of the two blocks for recovery. Specifically, given the global features $F_G$ and local features $F_L$, the aggregate process can be defined as:

$$F_A = F_G \oplus W F_L \quad (11)$$

where $W$ represents the learnable parameters, directly optimized by backpropagation and initialized as 1. It's worth noting that our design is exceptionally lightweight, as it does not introduce additional convolution layers. Finally, to capture richer and more accurate information, we refine the aggregated features $F_A$ to obtain the output feature $X_l$ of the ALGBlock as follows:

$$X_l = F_A \oplus f_{1\times1}^c(SG(f_{1\times1}^c(LN(X_{l-1} \oplus F_A)))) \quad (12)$$

## 4 EXPERIMENTS

We first describe the experimental details of the proposed ALGNet. Then we present both qualitative and quantitative comparisons between ALGNet and other state-of-the-art methods. Following that, we conduct ablation studies to validate the effectiveness of our approach. Finally, we assess the resource efficiency of ALGNet. Due to the page limit, additional results are provided in the supplementary material.

## 4.1 Experimental Settings

*4.1.1* ***Datasets****.* **Image Motion Deblurring.** Following recent methods [8, 47], we train ALGNet using the GoPro dataset [34], which includes 2,103 image pairs for training and 1,111 pairs for evaluation. To assess the generalizability of our approach, we directly apply the GoPro-trained model to the test images of the HIDE [37] and RealBlur [36] datasets. The HIDE dataset contains 2,025 images that collected for human-aware motion deblurring. Both the GoPro and HIDE datasets are synthetically generated, but the RealBlur dataset comprises image pairs captured under real-world conditions. This dataset includes two subsets: RealBlur-J, and RealBlur-R.

**Single-Image Defocus Deblurring.** To evaluate the effectiveness of our method, we adopt the DPDD dataset [1], following the methodology of recent approaches [8, 46]. This dataset comprises images from 500 indoor/outdoor scenes captured using a DSLR camera. Each scene consists of three defocused input images and a corresponding all-in-focus ground-truth image, labeled as the right view, left view, center view, and the all-in-focus ground truth. The DPDD dataset is partitioned into training, validation, and testing sets, comprising 350, 74, and 76 scenes, respectively. ALGNet is trained using the center view images as input, with loss values computed between outputs and corresponding ground-truth images.

*4.1.2* ***Training details****.* For various tasks, separate models are trained, and unless otherwise specified, the following parameters are utilized. The models are trained using the Adam optimizer [20] with parameters $\beta_1 = 0.9$ and $\beta_2 = 0.999$. The initial learning rate is set to $5 \times 10^{-4}$ and gradually reduced to $1 \times 10^{-7}$ using the cosine annealing strategy [30]. The batch size is chosen as 32, and patches of size $256 \times 256$ are extracted from training images. Data augmentation involves horizontal and vertical flips. We scale the network width by setting the number of channels to 32 and 64 for ALGNet and ALGNet-B, respectively.

## 4.2 Experimental Results

*4.2.1* ***Image Motion Deblurring****.* We present the performance of evaluated image deblurring approaches on the synthetic Go-Pro [34] and HIDE [37] datasets in Tables 1. Our ALGNet-B demonstrates a 0.43 dB improvement in performance over NAFNet-64 [4] on the GoPro [34] dataset. Compared with MambaIR [17], which is also based on the state space model, our ALGNet demonstrates an improvement in performance by 0.28 dB, while ALGNet-B achieves a substantial improvement of 0.84 dB. Additionally, as depicted in Figure 1, our ALGNet achieves even better performance through the scaling up of the model size, underscoring the scalability of ALGNet. Despite being trained solely on the GoPro [34] dataset, our network still achieves a significant gain of 0.19 dB PSNR over Restormer-Local [46] on the HIDE [37] dataset, demonstrating its generalization capability. Figure 4 illustrates that our model produces visually more pleasing results.

We also evaluate our ALGNet on real-world images from the RealBlur dataset [36] under two experimental settings: (1) applying the GoPro-trained model directly on RealBlur, and (2) training and testing on RealBlur data. As shown in Table 2, for setting 1, our ALGNet achieves performance gains of 0.16 dB on the RealBlur-R

**Table 1: Quantitative evaluations of the proposed approach against state-of-the-art motion deblrrring methods. The best and second best scores are highlighted and underlined. Our ALGNet-B and ALGNet are trained only on the GoPro dataset.**

| Methods | GoPro [34] PSNR ↑ | GoPro [34] SSIM ↑ | HIDE [37] PSNR ↑ | HIDE [37] SSIM ↑ |
|---|---|---|---|---|
| DeblurGAN-v2 [23] | 29.55 | 0.934 | 26.61 | 0.875 |
| MPRNet [47] | 32.66 | 0.959 | 30.96 | 0.939 |
| MPRNet-local [47] | 33.31 | 0.964 | 31.19 | 0.945 |
| HINet [5] | 32.71 | 0.959 | 30.32 | 0.932 |
| HINet-local [5] | 33.08 | 0.962 | - | - |
| Uformer [45] | 32.97 | 0.967 | 30.83 | **0.952** |
| MSFS-Net [50] | 32.73 | 0.959 | 31.05 | 0.941 |
| MSFS-Net-local [50] | 33.46 | 0.964 | 31.30 | 0.943 |
| NAFNet-32 [4] | 32.83 | 0.960 | - | - |
| NAFNet-64 [4] | 33.62 | 0.967 | - | - |
| Restormer [46] | 32.92 | 0.961 | 31.22 | 0.942 |
| Restormer-local [46] | 33.57 | 0.966 | 31.49 | 0.945 |
| IRNeXt [9] | 33.16 | 0.962 | - | - |
| SFNet [10] | 33.27 | 0.963 | 31.10 | 0.941 |
| FSNet [8] | 33.29 | 0.963 | 31.05 | 0.941 |
| DeblurDiNAT-S [28] | 32.85 | 0.961 | 30.65 | 0.936 |
| DeblurDiNAT-L [28] | 33.42 | 0.965 | 31.28 | 0.943 |
| MambaIR [17] | 33.21 | 0.962 | 31.01 | 0.939 |
| **ALGNet(Ours)** | 33.49 | 0.964 | 31.64 | 0.947 |
| **ALGNet-B(Ours)** | **34.05** | **0.969** | **31.68** | **0.952** |

**Table 2: Quantitative real-world deblurring results under two different settings: 1). applying our GoPro trained model directly on the RealBlur dataset [36], 2). Training and testing on RealBlur data where methods are denoted with symbol∗**

| Methods | RealBlur-R PSNR ↑ | RealBlur-R SSIM ↑ | RealBlur-J PSNR ↑ | RealBlur-J SSIM ↑ |
|---|---|---|---|---|
| MPRNet [47] | 35.99 | 0.952 | 28.70 | 0.873 |
| Restormer [46] | 36.19 | 0.957 | 28.96 | 0.879 |
| Stripformer [43] | 36.08 | 0.954 | 28.82 | 0.876 |
| FFTformer [21] | 35.87 | 0.953 | 27.75 | 0.853 |
| MRLPFNet [11] | 36.16 | 0.955 | 28.98 | 0.861 |
| DeblurDiNAT-S [28] | 35.92 | 0.954 | 28.80 | 0.877 |
| DeblurDiNAT-L [28] | 36.07 | 0.956 | 28.99 | 0.885 |
| MambaIR [17] | 35.98 | 0.955 | 28.82 | 0.875 |
| **ALGNet(Ours)** | **36.35** | **0.961** | **29.12** | **0.886** |
| DeblurGAN-v2∗ [23] | 36.44 | 0.935 | 29.69 | 0.870 |
| MPRNet∗ [47] | 39.31 | 0.972 | 31.76 | 0.922 |
| Stripformer∗ [43] | 39.84 | 0.975 | 32.48 | 0.929 |
| FFTformer∗ [21] | 40.11 | 0.973 | 32.62 | 0.932 |
| MRLPFNet∗ [11] | 40.92 | 0.975 | **33.19** | 0.936 |
| MambaIR∗ [17] | 39.92 | 0.972 | 32.44 | 0.928 |
| **ALGNet∗(Ours)** | **41.16** | **0.981** | 32.94 | **0.946** |

subset over Restormer [46] and 0.13 dB on the RealBlur-J subset over DeBlurDiNAT-L [28]. Compared with MambaIR [17], our gains are 0.37 dB and 0.30 dB on RealBlur-R and RealBlur-J, respectively.

A similar trend is observed for setting 2, where our gains over MRLPFNet [11] are 0.24 dB on RealBlur-R. Although our ALGNet performs slightly inferiorly to MRLPNet in PSNR metric on the RealBlur-J dataset, our SSIM metric is higher. Moreover, for setting 1, our method outperforms MRLPNet, indicating superior generalization capability. Figure 5 presents visual comparisons of the evaluated approaches. Overall, the images restored by our model exhibit sharper details and are closer to the ground truth compared to those produced by other methods.

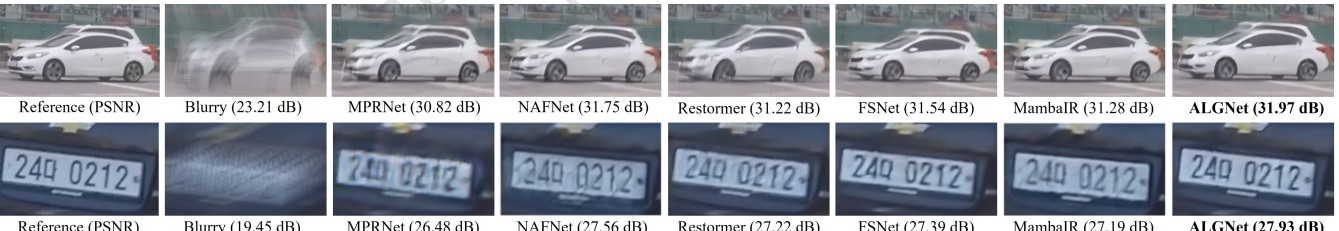

Reference (PSNR) | Blurry (23.21 dB) | MPRNet (30.82 dB) | NAFNet (31.75 dB) | Restormer (31.22 dB) | FSNet (31.54 dB) | MambaIR (31.28 dB) | **ALGNet (31.97 dB)**

Reference (PSNR) | Blurry (19.45 dB) | MPRNet (26.48 dB) | NAFNet (27.56 dB) | Restormer (27.22 dB) | FSNet (27.39 dB) | MambaIR (27.19 dB) | **ALGNet (27.93 dB)**

**Figure 4: Image motion deblurring comparisons on the GoPro dataset [34]. Our ALGNet recovers perceptually faithful images.**

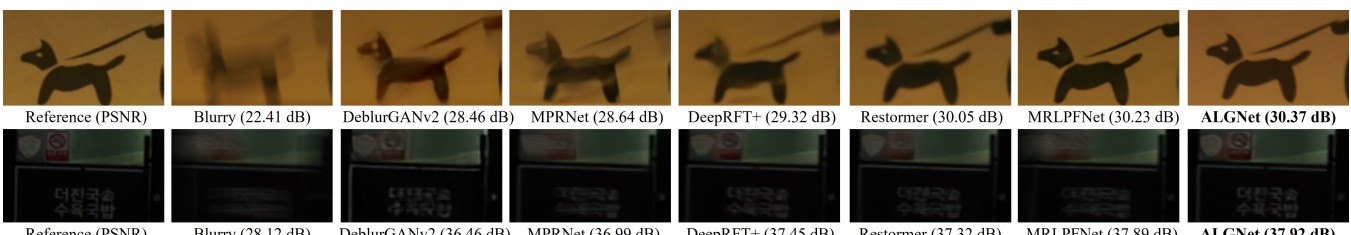

Reference (PSNR) | Blurry (22.41 dB) | DeblurGANv2 (28.46 dB) | MPRNet (28.64 dB) | DeepRFT+ (29.32 dB) | Restormer (30.05 dB) | MRLPFNet (30.23 dB) | **ALGNet (30.37 dB)**

Reference (PSNR) | Blurry (28.12 dB) | DeblurGANv2 (36.46 dB) | MPRNet (36.99 dB) | DeepRFT+ (37.45 dB) | Restormer (37.32 dB) | MRLPFNet (37.89 dB) | **ALGNet (37.92 dB)**

**Figure 5: Image motion deblurring comparisons on the RealBlur dataset [36]. Our ALGNet recovers image with clearer details.**

**Table 3: Quantitative comparisons with other single-image defocus deblurring methods on the DPDD testset [1] (containing 37 indoor and 39 outdoor scenes).**

| Methods | Indoor Scenes | | | | Outdoor Scenes | | | | Combined | | | |
|---|---|---|---|---|---|---|---|---|---|---|---|---|
| | PSNR ↑ | SSIM ↑ | MAE ↓ | LPIPS ↓ | PSNR ↑ | SSIM ↑ | MAE ↓ | LPIPS ↓ | PSNR ↑ | SSIM ↑ | MAE ↓ | LPIPS ↓ |
| EBDB [19] | 25.77 | 0.772 | 0.040 | 0.297 | 21.25 | 0.599 | 0.058 | 0.373 | 23.45 | 0.683 | 0.049 | 0.336 |
| DMENet [24] | 25.50 | 0.788 | 0.038 | 0.298 | 21.43 | 0.644 | 0.063 | 0.397 | 23.41 | 0.714 | 0.051 | 0.349 |
| JNB [38] | 26.73 | 0.828 | 0.031 | 0.273 | 21.10 | 0.608 | 0.064 | 0.355 | 23.84 | 0.715 | 0.048 | 0.315 |
| DPDNet [1] | 26.54 | 0.816 | 0.031 | 0.239 | 22.25 | 0.682 | 0.056 | 0.313 | 24.34 | 0.747 | 0.044 | 0.277 |
| KPAC [42] | 27.97 | 0.852 | 0.026 | 0.182 | 22.62 | 0.701 | 0.053 | 0.269 | 25.22 | 0.774 | 0.040 | 0.227 |
| IFAN [25] | 28.11 | 0.861 | 0.026 | 0.179 | 22.76 | 0.720 | 0.052 | 0.254 | 25.37 | 0.789 | 0.039 | 0.217 |
| Restormer [46] | 28.87 | 0.882 | 0.025 | **0.145** | 23.24 | 0.743 | 0.050 | **0.209** | 25.98 | 0.811 | 0.038 | **0.178** |
| IRNeXt [9] | 29.22 | 0.879 | 0.024 | 0.167 | 23.53 | 0.752 | 0.049 | 0.244 | 26.30 | 0.814 | 0.037 | 0.206 |
| SFNet [10] | 29.16 | 0.878 | **0.023** | 0.168 | 23.45 | 0.747 | 0.049 | 0.244 | 26.23 | 0.811 | 0.037 | 0.207 |
| FSNet [8] | 29.14 | 0.878 | 0.024 | 0.166 | 23.45 | 0.747 | 0.050 | 0.246 | 26.22 | 0.811 | 0.037 | 0.207 |
| MambaIR [17] | 28.89 | 0.879 | 0.026 | 0.171 | 23.36 | 0.738 | 0.051 | 0.243 | 26.11 | 0.809 | 0.039 | 0.202 |
| **ALGNet(Ours)** | **29.37** | **0.898** | **0.023** | 0.147 | **23.68** | **0.755** | 0.048 | 0.223 | **26.45** | **0.821** | **0.036** | 0.186 |

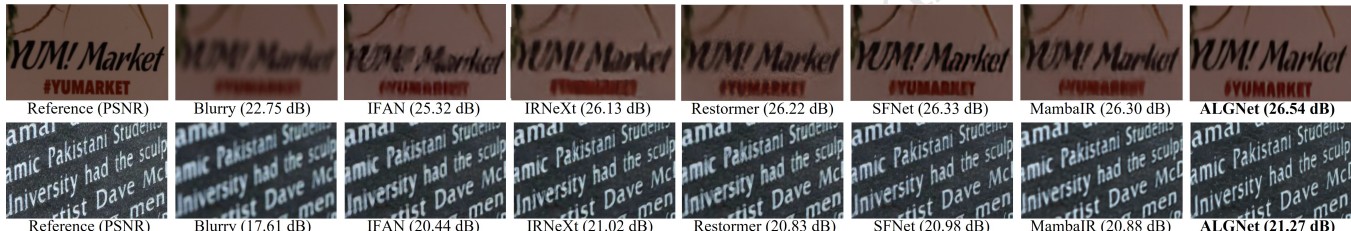

Reference (PSNR)    Blurry (22.75 dB)    IFAN (25.32 dB)    IRNeXt (26.13 dB)    Restormer (26.22 dB)    SFNet (26.33 dB)    MambaIR (26.30 dB)    **ALGNet (26.54 dB)**

Reference (PSNR)    Blurry (17.61 dB)    IFAN (20.44 dB)    IRNeXt (21.02 dB)    Restormer (20.83 dB)    SFNet (20.98 dB)    MambaIR (20.88 dB)    **ALGNet (21.27 dB)**

**Figure 6: Single image defocus deblurring comparisons on the DDPD dataset [1]. Our ALGNet effectively removes blur.**

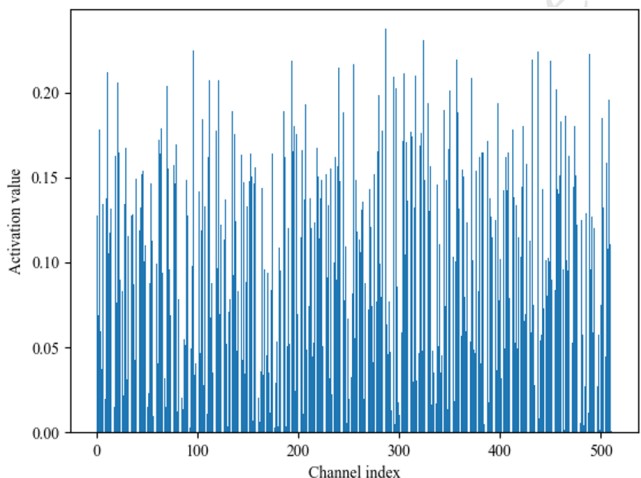

**Figure 7: The outputs of the CLGF module are processed by ReLU and global average pooling to obtain channel activation values.**

*4.2.2 **Single-Image Defocus Deblurring**.* We conduct single-image defocus deblurring experiments on the DPDD [1] dataset. Table 3 presents image fidelity scores of state-of-the-art defocus deblurring methods. ALGNet outperforms other state-of-the-art methods across all scene categories. Notably, in the combined scenes

category, ALGNet exhibits a 0.15 dB improvement over the leading method IRNeXt [9]. In comparison to MambaIR [17], which also relies on the state space model, our ALGNet showcases an improvement in performance by 0.48 dB in indoor scenes. The visual results in Figure 6 illustrate that our method recovers more details and visually aligns more closely with the ground truth compared to other algorithms.

### 4.3 Ablation Studies

Here we present ablation experiments to verify the effectiveness and scalability of our method. Evaluation is performed on the GoPro dataset [34], and the results are shown in Table. 4. The baseline is NAFNet [4]. We perform the break-down ablation by applying the proposed modules to the baseline successively, we can make the following observations:

(1) When our CLGF module consists of only one local branch or global branch, the improvement in deblurring performance is not significant. There are two main reasons for this. Firstly, our baseline model is already capable of fully capturing local information, rendering the addition of a local branch ineffective in enhancing the model's representation ability. Secondly, since our global branch is based on the state-space model, although it can capture long-distance information, it often encounters issues such as local pixel loss and channel redundancy, as illustrated in Figure 3. Therefore, when used alone, it fails to enhance performance.

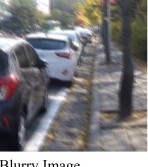 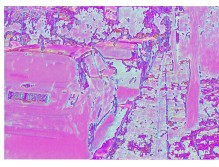 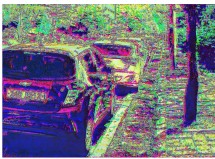 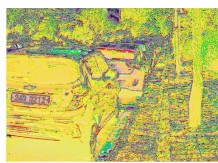 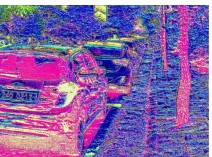 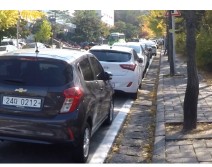

| Blurry Image | Initial Feature | Local-branch Feature | Global-branch Feature | CLGF+FA | Sharp Image |

**Figure 8: The internal features of ALGBlock. With our CLGF and FA, ALGBlcok produces more fine details than the initial feature, e.g., the number plate. Zoom in for the best view..**

**Table 4: Ablation study on individual components of the proposed ALGNet.**

| Method | PSNR |
|---|---|
| Baseline | 32.83 |
| Baseline + CLGF w/o local branch | 32.89 |
| Baseline + CLGF w/o global branch | 32.86 |
| Baseline + CLGF | 33.35 |
| Baseline + CLGF + FA | 33.49 |

**Table 5: The impact of feature aggregation method on the overall performance.**

| Modules | Sum | Concatenation | FA |
|---|---|---|---|
| PSNR | 33.35 | 33.37 | 33.49 |
| FLOPs(G) | 17 | 22 | 17 |

(2) When our CLGF module comprises both a local branch and a global branch, we observe a significant improvement in performance, up to 0.52 dB compared to the baseline. This indicates that CLGF has the ability to capture long-range dependency features and model local connectivity effectively.

(3) The FA contributes a gain of 0.14 dB to our model.

To further validate the effectiveness of our CLGF module, we apply ReLU and global average pooling operations on the output results of CLGF to obtain channel activation values (see Figure 7). It's evident that our CLGF successfully circumvents the issue of channel redundancy caused by an excessive number of hidden states in the state spaces model.

Furthermore, to assess the advantage of our FA design, we compare it with other methods such as sum and concatenation. As shown in Table 5, our FA consistently delivers superior results, indicating its effectiveness in emphasizing the importance of the local branch in restoration. Importantly, our design does not introduce any additional computational burden.

Finally, we compare the feature maps before and after our ALGBLock in Figure 8. It is evident that the feature map from our local branch contains more detailed information compared to that from the global branch. Upon aggregation of the local and global branches of CLGF using FA, we observe a significant recovery of more details, particularly in the blurred license plate number present in the initial feature map.

**Table 6: The evaluation of model computational complexity on the GoPro dataset [34]. The FLOPs are evaluated on image patches with the size of 256×256 pixels. The running time is evaluated on images with the size of 1280 × 720 pixels.**

| Method | Time(s) | FLOPs(G) | PSNR | SSIM |
|---|---|---|---|---|
| MPRNet [47] | 1.148 | 777 | 32.66 | 0.959 |
| Restormer [46] | 1.218 | 140 | 32.92 | 0.961 |
| Stripformer [43] | 1.054 | 170 | 33.08 | 0.962 |
| IRNeXt [9] | 0.255 | 114 | 33.16 | 0.962 |
| SFNet [10] | 0.408 | 125 | 33.27 | 0.963 |
| FSNet [8] | 0.362 | 111 | 33.29 | 0.963 |
| MambaIR [17] | 0.743 | 439 | 33.21 | 0.962 |
| ALGNet(Ours) | **0.237** | **17** | **33.49** | **0.964** |

## 4.4 Resource Efficient

We assess the model complexity of our proposed approach and state-of-the-art methods in terms of model running time and FLOPs. Table 6 and Figure 1 illustrate that our ALGNet model achieves SOTA performance while simultaneously reducing computational costs. Specifically, we achieve a 0.2 dB improvement over the previous best approach, FSNet [8], with up to 84.7% cost reduction and nearly 1.5 times faster inference. Compared to MambaIR [17], our ALGNet reduces computational costs by 96.1% and achieves 3.1 times faster inference. This underscores the efficiency of our method, demonstrating superior performance along with resource effectiveness.

## 5 CONCLUSION

In this paper, we propose an efficient image deblurring network that leverages selective state spaces model to aggregate enriched and accurate features. We design an ALGBlock consisting of CLGF and FA module. The CLGF module captures long-range dependency features using a selective state spaces model in the global branch, while employing simplified channel attention to model local connectivity in the local branch, thus reducing local pixel forgetting and channel redundancy. Additionally, we propose the FA module to emphasize the significance of the local information by dynamically recalibrating the weights through a learnable factor during the aggregation of the CLGF two branches. Experimental results demonstrate that the proposed method outperforms state-of-the-art approaches.

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
