# OpenReview forum: "Learning Enriched Features via Selective State Spaces Model for Efficient Image Deblurring"
_acmmm.org/ACMMM/2024/Conference — MM2024 Poster_

### Official Review · Reviewer_K25P · 2024-05-17

**Rating:** 3
**Confidence:** 3

**Summary:**

This paper introduces the selective state space model, renowned for its long-range modeling within linear complexity, into the image deblurring task, further developing an aggregate local and global information block. Experimental results demonstrate that the proposed algorithm outperforms state-of-the-art approaches on various benchmarks.

**Strengths:**

1. The proposed approach conduct the complementary global and local branch, implemented by a selective state space model and channel attention.

2. The proposed ALGNet achieves the superior performance while preserving efficiency.

**Limitations:**

1. MambaIR designs a Residue State Space Block by cascading VSSM and channel attention. It seem that the ALGNet reformulates this cascading structure into a parallel one. The authors are encouraged to discuss the specific improvements brought by this reformulation, excluding the performance gains.

2. The authors claim that this combination of global and local brancher not only addresses issues like local pixel forgetting and channel redundancy but also empowers the network to capture more enriched and precise features. However, Figure 3 only illustrates the channel redundancy of the global branch. It does not adequately demonstrate how introducing the local branch alleviates this problem. Additional evidence is needed to support this claim.

3. The reviewer is curious whether cascading global and local branches can achieve comparable performance and diverse channel responses. This alternative structure should be explored and compared with the proposed parallel design.

4. The compared baselines in Tables 1, 2, and 4 are inconsistent. The authors are suggested to include the missing baselines in Tables 2 and 4. Specifically, Table 4 lacks comparisons with efficient algorithms like FFTformer [1]. Including such comparisons would provide a more comprehensive evaluation of the proposed method’s performance and efficiency.

[1] Kong L, Dong J, Ge J, et al. Efficient frequency domain-based transformers for high-quality image deblurring[C]//Proceedings of the IEEE/CVF Conference on Computer Vision and Pattern Recognition. 2023: 5886-5895.

**Suitability:**

2

---

### Official Review · Reviewer_sSQa · 2024-05-24

**Rating:** 3
**Confidence:** 4

**Summary:**

This paper introduces an efficient image deblurring network, ALGNet, which utilizes a selective state space model to aggregate enriched features. The network consists of multiple ALGBlocks, each containing a capturing local and global features module (CLGF) and a feature aggregation module (FA). Experimental results demonstrate that ALGNet outperforms state-of-the-art methods on multiple benchmarks while maintaining high computational efficiency.

**Strengths:**

1. The innovation of this paper lies in the proposal of a novel ALGBlock, which integrates the selective state space model and channel attention mechanism. The model outperforms existing methods in multiple benchmark tests while maintaining high computational efficiency.
2. The paper provides clear technical details to ensure the reproducibility of the experiments. The content is clearly expressed, and the figures effectively demonstrate the model structure and experimental results.
3. The ablation experiment design is comprehensive, effectively verifying the role of individual components.

**Limitations:**

1. The novelty of the manuscript is somewhat week. The proposed method combines convolution and SSM. However, the idea of combining local processing and long-term dependency is commonly-seen in the transformer-based methods. Compared to these methods, the contribution of this method mainly lieds in the replacement of SSM with self-attention, which is somewhat week.
2. The related RNN-based works should be discussed. By viewing image as pixel/patch sequence, NLP architecutre can be used for image processing. Except for transformer, RNN has also been considered for image restoration, see [1,2]. The discussion about the advantages of SSM over RNN is necessary here.
   [1] Learning Recursive Filters for Low-Level Vision via a Hybrid Neural Network,ECCV,2016
   [2] Deblurring Dynamic Scenes via Spatially Varying Recurrent Neural Networks, TPAMI, 2021
3. As I know, one of the advantages of SSM over RNN is that SSM allows parallel training. So, It would be better to highlight the advantage by comparing the training efficiency. However, relative experiments are lack in the manuscript.

**Suitability:**

2

---

### Official Review · Reviewer_sTnW · 2024-05-24

**Rating:** 3
**Confidence:** 3

**Summary:**

This paper propose an UNet-strctured network with designed ALGBlocks for image deblurring. ALGBlock uses global branches with Mamba to realize linear-complexity long-range dependencies, and uses local branch to address pixel forgetting caused by Mamba. Subsequently it uses weighted summation for branch feature aggregation and finally uses refined blocks to obtain the output of block. Experimental results
show it outperforms existing approaches.

**Strengths:**

1. The calirity is generally good.
2. The technical seems correct. The idea is clear.
3. The experiments contains Image Motion Deblurring and Single-Image Defocus Deblurring tasks, covering 4 datasets overall, which can show the generalization of the proposed method.

**Limitations:**

1. Each subtitle in Related Works is still underconsidered. In my view, they should not be divided into 4 parts. They are 4 types of strategies the image deblurring adopts.
2. The novelty of overall method seems marginal. The overall architecture is UNet. The multi-input and multi-output is not a new strategy. Although the two-branch CLGF is issuse-driven, it is designed all based on previous works.
3. How to acquire the multiple inputs and outputs are not given in the text.
4. The writing logical of section 3.1 is a little confused. In line 404, the top branch (local branch) is mentioned first, but it is mentioned secondly in line 481 after introfucing the bottom branch (glocal branch). Also, there is a writing mistake in line 477 "as per [17]".
5. Although experimental results of several datasets prove the effectivenesss of proposed model, the improvement is minor.
6. It would be more convincing by showing the experimental result to prove that the local branch could solve the problem of pixel forgetting.

**Suitability:**

2

---

### Official Review · Reviewer_W2q2 · 2024-05-25

**Rating:** 5
**Confidence:** 3

**Summary:**

This paper propose a new network architecture called ALGNet for image deblurring, which aims to efficiently aggregate local and global features.ALGNet has designed a block that integrates local and global information (ALGBlock), which includes a module that captures local and global features (CLGF) and a feature aggregation module (FA). Experimental results show that ALGNet performs better than existing state-of-the-art methods while maintaining computational efficiency.

**Strengths:**

1 The use of selective state spaces model (SSM) and the combination of local and global feature capturing modules (CLGF) is a novel approach that addresses the limitations of previous methods, such as local pixel forgetting and channel redundancy.

2 The FA module effectively highlights local features, which are crucial for image deblurring, without introducing significant computational overhead.

3 The paper presents extensive qualitative and quantitative evaluations on multiple datasets, including GoPro, HIDE, and RealBlur, demonstrating the generalizability and robustness of ALGNet.

4 ALGNet achieves significant improvements in performance while reducing computational costs and inference time, making it suitable for real-world applications.

**Limitations:**

1 The selective state spaces model (SSM) used in the global branch tends to neglect local pixel information due to the flattening strategy and extensive distances between adjacent pixels when processed in a 1D sequence. This can result in a loss of local details that are crucial for effective image deblurring.

2 Although the SSM aims to model long-range dependencies with linear complexity, the four-directional scanning approach and the state spaces computations lead to increased computational overhead. This can potentially sacrifice the advantage of low computational resource utilization, which is one of the main benefits of using state space models.

3 The paper mentions that image deblurring often deals with high-resolution images, but the experimental results do not explicitly show the performance on such images. Including experiments on high-resolution images would help in understanding how well the model scales and performs on larger images.

**Suitability:**

2

---

### Meta-Review · Area_Chair_xuPT · 2024-07-02

**Recommendation:** Accept (Poster)
**Confidence:** 5

**Metareview:**

In this paper, the authors propose an image deblurring network (ALGNet), which applies the selective state space model to aggregate the enriched features. The multiple ALGBlocks include a local and global feature capturing module and a feature aggregation module. The results show that the proposed ALGNet exhibits a relative better performance on various benchmarks than SOTA methods, while maintaining high computational efficiency. However, the reviewers still concern its weak novelty.

After rebuttal, this paper receives 3 positive reviews and 1 negative review, so I recommend the acceptance of this paper.